# Direct Antiviral Treatments for Hepatitis C Virus Have Off-Target Effects of Oncologic Relevance in Hepatocellular Carcinoma

**DOI:** 10.3390/cancers12092674

**Published:** 2020-09-19

**Authors:** Catia Giovannini, Francesca Fornari, Valentina Indio, Davide Trerè, Matteo Renzulli, Francesco Vasuri, Matteo Cescon, Matteo Ravaioli, Alessia Perrucci, Annalisa Astolfi, Fabio Piscaglia, Laura Gramantieri

**Affiliations:** 1Center for Applied Biomedical Research (CRBA), Azienda Ospedaliero-Universitaria di Bologna, 40138 Bologna, Italy; catia.giovannini4@unibo.it (C.G.); alessia.perrucci@studio.unibo.it (A.P.); 2Department of Medical and Surgical Sciences, University of Bologna, 40138 Bologna, Italy; matteo.cescon@unibo.it (M.C.); fabio.piscaglia@unibo.it (F.P.); 3“Giorgio Prodi” Cancer Research Center (CIRC), University of Bologna, 40138 Bologna, Italy; valentina.indio2@unibo.it (V.I.); annalisa.astolfi@unibo.it (A.A.); 4Program of Laboratory Medicine, Azienda Ospedaliero-Universitaria di Bologna and Department of Experimental, Diagnostic and Specialty Medicine, University of Bologna, 40138 Bologna, Italy; davide.trere@unibo.it; 5Radiology Unit, Azienda Ospedaliero-Universitaria di Bologna, 40138 Bologna, Italy; matteo.renzulli@aosp.bo.it; 6Pathology Unit, Azienda Ospedaliero-Universitaria di Bologna, 40138 Bologna, Italy; francesco.vasuri@aosp.bo.it; 7Department of Surgery, Azienda Ospedaliero-Universitaria di Bologna, 40138 Bologna, Italy; matteo.ravaioli@aosp.bo.it; 8Division of Internal Medicine Unit, Azienda Ospedaliero-Universitaria di Bologna, 40138 Bologna, Italy

**Keywords:** hepatocellular carcinoma (HCC), hepatitis C virus (HCV), direct-acting antiviral agents (DAAs)

## Abstract

**Simple Summary:**

Hepatitis C virus (HCV) eradication by direct-acting antiviral agents (DAAs) reduces de novo hepatocellular carcinoma incidence in cirrhosis; however, contrasting evidence on higher incidence of hepatocellular carcinoma (HCC) was reported in patients previously treated for HCC. Here, we showed that sofosbuvir and daclatasvir can modulate cell proliferation, invasion capability and gene expression in HCC-derived cell lines, suggesting that off-target effects of these drugs might be responsible for both the increase and reduction of cell proliferation and migration capability. Off-target gene modulation, mainly affecting mitochondrial functions, ribosomal genes and histones, was consistent with matched phenotypic changes and might account either for pro-oncogenic or tumor-suppressive functions of DAAs, that seemed to be dictated by the molecular background.

**Abstract:**

Background and Aims: HCV eradication by direct-acting antiviral agents (DAAs) reduces de novo hepatocellular carcinoma (HCC) incidence in cirrhosis; however, contrasting evidence about beneficial or detrimental effects still exists in patients who have already developed HCC. Methods: we investigated whether sofosbuvir and daclatasvir modulate cell proliferation, invasion capability and gene expression (RNA-seq) in HCC-derived cell lines, hypothesizing possible off-target effects of these drugs. Results observed in HCC cell lines were validated in non-HCC cancer-derived cell lines and a preliminary series of human HCC tissues by qPCR and IHC. Results: DAAs can affect HCC cell proliferation and migration capability by either increasing or reducing them, showing transcriptomic changes consistent with some unexpected drug-associated effects. Off-target gene modulation, mainly affecting ribosomal genes, mitochondrial functions and histones, points to epigenetics and proliferation as relevant events, consistent with matched phenotypic changes. A preliminary validation of in vitro findings was performed in a restricted cohort of HCC patients previously treated with DAAs, with immunohistochemical correlations suggesting DAA-treated HCCs to be more aggressive in terms of migration and epidermal-to-mesenchymal transition. Conclusions: Our findings suggested the possible occurrence of off-target effects ultimately modulating cell proliferation and/or migration and potentially justified previous findings showing some instances of particularly aggressive HCC recurrence as well as reduced incidence of recurrence of HCC following treatment with DAAs.

## 1. Introduction

In western countries as well as in Japan, a large proportion of hepatocellular carcinomas (HCCs) are associated with hepatitis C virus (HCV) infection in the cirrhotic stage. It was hardly possible to cure this infection in the cirrhotic phase until the advent of direct-acting antiviral agents (DAAs). The target mechanism of action is inhibition of viral replication by blocking non-structural viral proteins. Therefore, they are classified as NS3-4A inhibitors, NS5A inhibitors and nucleoside or non-nucleoside inhibitors of NS5B, according to the viral protein they block. A significant reduction in first HCC incidence results from HCV eradication by DAAs [1]. However, concern still exists about treatment of HCV infection in cirrhotic patients at risk of HCC recurrence. In fact, some studies claimed an accelerated HCC recurrence after DAAs [2,3,4], whereas others did not confirm these findings, but rather showed a reduction in HCC incidence [5,6,7]. However, even if we did not consider any increase in HCC recurrence, some studies showed a more aggressive HCC pattern in those who recurred [8]. The hypothesis of some influence of DAAs on tumor development or tumor inhibition is reinforced by the report of occurrence of other malignancies in patients treated by DAAs [9,10,11], including cholangiocarcinoma [12]. Unfortunately, no solid evidence has been produced, at a molecular level, supporting any influence of DAAs on tumor behavior.

One hypothesis is that DAAs do not induce neoplasia directly; rather, HCV clearance might reduce the immune response, favoring the development or progression of neoplastic clones, especially in patients with pre-existing dysplastic/neoplastic foci [8]. Faillaci et al. showed that DAA-mediated increase of VEGF could trigger neo-angiogenesis [13]. Besides these mechanisms, the mitochondrial toxicity of other nucleotide inhibitors should be considered, together with the inhibition of human mitochondrial DNA polymerase, which is a side effect of antiviral drugs targeting viral RNA-dependent RNA polymerase [14], opening the question as to whether off-target effects might be relevant in this specific context too.

Off-target effects of DAAs were previously described. Bachovchin et al. [15] used high-throughput serine hydrolase inhibitor screening and suggested that serious skin reactions induced by telaprevir, a HCV-NS3 serine protease inhibitor, might result from an off-target inhibition of CELA1, a serine hydrolase expressed in the skin. They also pointed to the fact that off-target effects apply to specific molecules/drugs, and cannot be generalized to the structurally similar drugs of the same class. Similarly, NS5A inhibitors, daclatasvir and ledipasvir, were reported to induce a dose-dependent DNA damage through the oxidation of both nucleobases and deoxyribose moieties of DNA [16].

Sofosbuvir and daclatasvir have been the most used DAAs in Italian cirrhotic patients. Daclatasvir inhibits NS5A. In clinical practice, it is usually combined with sofosbuvir, which is a nucleoside polymerase inhibitor of the RNA polymerase NS5B. After phosphorylation to nucleoside triphosphate, it competes with nucleotide substrates that are incorporated into the viral RNA chain, resulting in chain termination.

In order to screen for molecular mechanisms linking DAAs to tumor behavior, we tested whether possible off-target effects might occur in HCC-derived cell lines, by evaluating sofosbuvir and daclatasvir modulation of cell proliferation, invasion capability and gene expression. The results observed in HCC cell lines were validated in non-HCC cancer-derived cell lines and a restricted series of primary human HCCs. 

## 2. Methods

### 2.1. Cell Culture and Treatments

HepG2, Hep3B, SNU449 and SNU475 cell lines were obtained from the American Type Culture Collection (ATCC, Rockville, MD, USA) and were maintained according to ATCC instructions. Huh7 cells were from Prof. Alberti’s laboratory, University of Padua (Padua, Italy), and were cultured with RPMI-1640 (Life Technologies, Carlsbad, CA, USA). Hep10 cells were purchased from Thermo Fisher Scientific (Waltham, MA, USA). TFK1 and Huh28 cholangiocarcinoma cell lines were provided by Prof. Svegliati Baroni, University of Ancona (Ancona, Italy), and were grown in RPMI-1640. LnCAP and MCF7 cells, respectively, derived from prostate cancer and breast cancer, were obtained from the ATCC. Media were supplemented with 10% fetal bovine serum (FBS), L-glutamine and penicillin/streptomycin (Life Technologies).

Cell proliferation assay, cell cycle analysis, scratch tests, real-time qPCR and immunohistochemistry (IHC) on DAA-treated cells were performed as detailed in the Appendix A. 

### 2.2. RNA Sequencing and Bioinformatics Analysis

RNA sequencing was used to analyze the gene expression profiles of HCC-derived cell lines following DAA treatments as well as primary HCCs and cirrhotic tissues. Total RNA was extracted from cell lines (treated with DAAs for 24 h) and tissue samples of HCC and adjacent cirrhosis (control and next-generation sequencing (NGS) cohorts) by using TRIzol (Invitrogen, Life Technologies). Whole transcriptome sequencing was performed on the Illumina HiScanSQ platform by using the paired-end strategy (2 × 80 bases) as previously described [17], producing a total of 370 gigabases and an average of 95 × 10^6^ reads per sample with an average deep of coverage of ~60X. After BCL to FASTQ conversion (bcl2fastq—https://www.illumina.com), quality control and adapter trimming (AdapterRemoval—https://github.com/MikkelSchubert/adapterremoval), the paired-end reads were mapped on the human reference genome HG19 with TopHat/Bowtie (http://ccb.jhu.edu/software/tophat). The abundance of transcripts was computed by using the Python function htseq-count (https://htseq.readthedocs.io) with the Ensembl72 release as the gene feature annotation. Gene expression profiling was performed by using a suite of R-bioconductor packages (https://bioconductor.org), such as edgeR (for normalization as count per million) and limma (to perform the differential expression analysis).

Principal component analysis (PCA) was performed with prcomp function (R package stats) and the 3D projections were visualized using plot3d (R package ggplot2). All the heatmaps were built with the R package pheatmap. A Venn diagram analysis was performed to visualize the overlapping genes of the different clusters using an online software (http://bioinformatics.psb.ugent.be/webtools/Venn/).

### 2.3. Patients

Two independent patient cohorts were examined: 

The first one (post-DAA cohort) was composed of 13 patients with first HCC diagnosed and surgically resected within 12 months (range 0–11) from the end of DAA treatment (assuming that malignant cells were already present at the time of DAA treatment). Formalin-fixed, paraffin-embedded (FFPE) HCC and surrounding tissues were available; thus, these patients were studied by IHC to determine whether the deregulated expression of molecules in DAA-treated HCC cell lines might be confirmed in this subgroup of HCCs too.

The second cohort (active HCV cohort) was composed of 39 surgically resected HCC patients with active HCV infection and who were not previously exposed to DAAs. These patients were studied by IHC and RT-qPCR. Of these, 28 patients, who were surgically resected for first HCC in a background of active HCV infection in the pre-DAA era, had already been analyzed by RNA sequencing (RNA-seq), as previously reported [17]. HCC gene expression from this cohort was compared with gene expression profiles following in vitro DAA treatments, to ascertain common molecular changes. For validation purposes of NGS and in vitro data, qPCR analyses were performed in the active HCV cohort as well as in treated cell lines.

Characteristics of these patient cohorts are described in Table 1.

### 2.4. Statistical Analysis

For both patient samples and cell lines, the statistical methods adopted to evaluate significant differences in gene expression were as follows: differentially expressed genes were evaluated using the eBayes approach (R-bioconductor package limma, 3.12, Bioconductor, Wisconsin, WI, USA) for the HCC versus surrounding cirrhosis comparison (paired test) and treated versus untreated cell lines (unpaired test). Log2 fold change, nominal *p*-value and corrected *p*-value (*q*-value with Benjamini–Hochberg adjustment) were reported and genes significantly modulated were considered with *q*-value < 0.01. RNA-seq raw data are available upon request. For qPCR studies (human tissues and cell lines) and IHC studies, differences between groups were analyzed using a double-sided Student’s *t*-test. Pearson’s correlation coefficient was used to investigate relationships between two variables (qPCR gene expression studies). Differences between categorical variables were explored by Chi-square test. Reported *p*-values were two-sided and considered significant when lower than 0.05 (* *p* < 0.05, ** *p* < 0.01, *** *p* < 0.001, **** *p* < 0.0001). Statistical calculations were performed using SPSS version 19.0 (SPSS Inc, Armonk, NY, USA; IBM Corp, New York, NY, USA). Gene function and pathway analysis was performed on the differentially expressed genes using Visualization and Integrated Discovery v6.7 (http://david.abcc.ncifcrf.gov/home.jsp).

## 3. Results

### 3.1. Sofosbuvir and Daclatasvir Affect Proliferation of HCC Cell Lines 

Our first question was whether a single administration of sofosbuvir and/or daclatasvir plays any effect on HCC cell proliferation. We performed a preliminary screening of five HCC-derived cell lines (SNU449, SNU475, Huh7, HepG2 and Hep3B) by crystal violet staining after exposure to either sofosbuvir and/or daclatasvir. Proliferation was increased, reduced or unaffected at 24 h, depending on the cell line (Figure 1). Any effect disappeared at 48 h from drug exposure. 

To go deeper into the mechanisms sustaining variations of proliferation, we analyzed cell cycle after DAA treatments. Sofosbuvir reduced Huh7 and SNU449 cell proliferation by arresting cells in the G1 phase. A limited increase of proliferation was induced by sofosbuvir in SNU475 cells. Daclatasvir triggered cell proliferation resulting from a reduction of the G1 phase in SNU475 and SNU449 cells, while it had no effect on Huh7 cells. Either drug did not modify the proliferation or cell cycle of HepG2 and Hep3B cells (Figure 1). Neither daclatasvir nor sofosbuvir modified the proliferation of Hep10 cells, which remained in the G0 phase in more than 97% of cells.

These data showed that DAAs can modify HCC cell proliferation, either by reducing or increasing it, in the different cell lines and pointed to the relevance of the molecular background in dictating DAAs effects.

### 3.2. Sofosbuvir and Daclatasvir Affect Migration of HCC Cells 

Cancer cells may not proliferate while migrating, and they may stop moving while dividing. Hence, migration and proliferation may be mutually exclusive. Previous studies showed that microvascular invasion and aggressive features characterize HCCs that develop after DAAs [4,18]. Therefore, we determined the role of DAAs in cell movement. The Incucyte Live-Cell Analysis System revealed that sofosbuvir increased the ability of SNU449 and, to a lesser extent, Huh7 cells, to fill in the wound, whereas no influence was observed in SNU475, Hep3B and HepG2 cells. Conversely, daclatasvir affected the migration capability only in SNU475 cells (Figure 2). Hep10 cells were not tested for migration, since they were cultured in suspension. The selective activity of the study drugs in specific cell lines reinforced the concept that the molecular background (expected to correspond to individual patient situations) governs the effects of DAAs on cell migration as previously observed for proliferation. 

### 3.3. Sofosbuvir and Daclatasvir Affect Proliferation and Migration of Non-HCC Cancer Cell Lines

Based on our results in HCC cells, we investigated whether DAAs might also affect cell proliferation and migration of cholangiocarcinoma (CCA) cells. To this aim, TFK1 and Huh28 cells treated with sofosbuvir and daclatasvir were analyzed by crystal violet staining, flow cytometry (FACS) analysis and Incucyte Live-Cell Analysis. Sofosbuvir increased proliferation and accelerated cell cycle progression, displayed by G1-phase reduction, in both TFK1 and, to a lesser extent, Huh28 cells. Daclatasvir increased cell proliferation in both cell lines, although with less prominent variations of cell cycle (Appendix A). A wound healing assay showed that sofosbuvir enhanced the migratory capabilities of both CCA cell lines, while daclatasvir had no effect.

We additionally tested MCF7 (breast cancer-derived) and LnCAP (prostate cancer-derived) cell lines to study the effects of sofosbuvir and daclatasvir on extrahepatic cancer cells. Neither sofosbuvir nor daclatasvir affected MCF7 cell proliferation and migration. Conversely, LnCAP cells displayed an increase in proliferation after sofosbuvir treatment, resulting from G1-phase reduction (Appendix A), while the migration test was not fully reliable due to the growth pattern of these cells, which are weakly adherent and grow in aggregates. 

These data confirmed the influence of specific cellular backgrounds on DAA-mediated effects on the proliferation and migration of non-HCC cell lines. Thus, we can speculate that off-target effects of DAAs inducing either decreased or increased cell proliferation and migration depend upon the genetic backgrounds of some but not all tumor types and liver lineages, in keeping with the heterogeneous clinical HCC courses.

### 3.4. DAAs Modulate Gene Expression of HCC-Derived Cells

A gene expression analysis of DAA-treated cells was performed by RNA-seq in those HCC cell lines that displayed the most relevant variations of proliferation and/or migration, triggered by sofosbuvir and daclatasvir. In detail, gene expression was profiled in Huh7 cells treated with sofosbuvir, SNU475 cells treated with daclatasvir and SNU449 cells treated with both drugs. Each treated cell line was compared with its matched dimethyl sulfoxide (DMSO)-treated control. To account for variations among biological replicates, experiments were done in duplicate. Principal component analysis (PCA) of treated cells showed a clear separation between cell lines, suggesting distinct expression profiles linked to cell type and minor differences due to the treatments (Figure 3A). Looking to the union set of genes that were differentially expressed in each of the comparisons (adjusted *p*-value < 0.01, *n* = 4783), we were able to highlight substantial differences in expression patterns that also showed through the hierarchical clustering analysis (Appendix A). Assuming that the effects of the DAAs are dependent on both the specific drug and the cellular background, a Venn diagram was used to visualize the overlapping genes regulated by DAAs in treated cells (Appendix A). Gene expression was compared according to phenotypic changes induced by DAAs in each cell line.

Sofosbuvir reduced proliferation in SNU449 and Huh7 cells. Several commonly deregulated genes (*n* = 321) were identified, including 50 upregulated genes, 256 downregulated genes and 15 genes that were differently regulated in the two cell lines. Functional enrichment analyses showed that co-downregulated genes were significantly enriched in 20 Gene Ontology-Biological Process (GO-BP) terms, with the top 10 reported in Table 2. The 321 commonly deregulated genes were subjected to STRING (v.9.0) analyses to highlight functional interactions. STRING analyses, carried out at high stringency, evidenced two major interaction networks (Figure 3B). The first block contained histone proteins, while the second one included ribosomal proteins, highlighting epigenetics and ribosomal biogenesis and function as relevant events contributing to reduced proliferation. Of note, four gene families, co-downregulated in cells whose proliferation was reduced by sofosbuvir (oxidative phosphorylation, respiratory chain, electron transport and ATP biosynthetic process), were associated with mitochondrial functions. 

Since daclatasvir triggered proliferation in SNU449 and SNU475 cells, we looked for overlapping genes whose deregulation might affect cell proliferation in both cell lines. Sixty commonly deregulated genes were identified, including 59 upregulated genes and 1 (RN7SL3) downregulated gene, as shown in Appendix A. No significant GO-BP terms emerged from functional enrichment analyses even though deregulated gene families were mainly related to mitochondrial function, ribosomal proteins, small nuclear and small nucleolar RNAs and a group of genes with previously reported roles in cancer, including oncogenes HSPG2, LAMA5 and SCL25A2 [19], LAMA5 [20] and SCL25A29 [21].

To explore drug-dependent effects, we compared gene expression in SNU449 cells treated by either of the DAAs. In this cell line, daclatasvir triggered proliferation without affecting migration, while sofosbuvir reduced proliferation, while increasing migration capability. Several commonly deregulated genes and pseudogenes (*n* = 101), upregulated by daclatasvir and downregulated by sofosbuvir, were identified (Table 3), which could be categorized into four GO-BP clusters, again representing transcription regulation, mitochondria-related genes and ribosomal protein-coding genes. This opposite modulation was consistent with opposite effects on proliferation and, possibly, on migration. Resulting proteins encoded by genes reported in Table 2 were subjected to a high-stringency STRING (v.9.0) analysis, which highlighted reciprocal interactions between 18 proteins. A cluster of proteins clearly described by functional interactions consisted of seven anti-apoptotic components of the humanin-like family (Figure 3C) and mitochondria-derived peptides, exhibiting strong cytoprotective actions against stressing events.

We finally compared gene expression changes in cell lines that displayed increased cell migration (i.e., SNU475 cells treated with daclatasvir and SNU449 and Huh7 cells treated with sofosbuvir). Twenty-seven common genes were identified (Table 4), among which we again recognized members of the humanin-like family, ribosomal protein subunit-coding genes, small nucleolar and small nuclear RNAs, histone-coding genes and tumor-associated genes such as LAMA5 [20] and HSPG2 (perlecan) involved in cell adhesion and pathologic angiogenesis [19].

These findings supported a role of sofosbuvir and daclatasvir as triggers of gene expression changes in HCC cells. Remarkably, these changes were in accordance with corresponding proliferation and migration changes observed in vitro. 

Among these deregulated genes, expression changes of two histones (HIST1H4A and HIST1H3A) were validated by qPCR. Both the histone-coding genes were downregulated in primary HCCs, which suffered post-surgical recurrence within two years after surgery (Student’s *t*-test, *p* < 0.05; Figure 4A–C). In keeping with these findings, HCC cell lines with increased migration capability induced by DAAs (SNU475 cells treated by daclatasvir and SNU449 cells treated by sofosbuvir) displayed reduced expression of both histones, mirroring primary HCCs with recurrent disease after surgery (Figure 4D,E).

### 3.5. DAA Treatment Regulates HCC-Related Genes

To support the clinical relevance of in vitro findings, we scrutinized for genes that were consistently deregulated in DAA-exposed cell lines and in human HCC tissues (compared with the surrounding paired cirrhosis) that were not treated with DAAs. This analysis was expected to indicate a possible relevance in human hepatocarcinogenesis of genes deregulated in vitro by DAAs. We focused on genes with an annotated function in cancer. This procedure led to the identification of 90 genes that were deregulated in vitro and ex vivo. The 37 genes involved in proliferation were oppositely deregulated in HCC tissues and in the two cell lines (Huh7 and SNU449) whose proliferation was reduced by sofosbuvir. Conversely, the 25 genes involved in migration were upregulated in the same cell lines, which showed accelerated migration after sofosbuvir and in HCC tissues as well. The corresponding numbers for daclatasvir (tested in SNU475 and SNU449 with increased proliferation and which displayed different effects on migration) were 15 genes associated with proliferation and 13 genes related to migration (Figure 4F).

### 3.6. Immunohistochemistry Evaluation in HCC with and without DAAs

To further assess the extent to which in vitro findings were representative of HCCs developed after DAA treatment, we analyzed by IHC the expression of Forkhead box protein M1 (FOXM1), Ki67 and vimentin (VIM) in thirteen surgically resected HCCs diagnosed within 12 months after completion of treatment with DAAs (post-DAA cohort) and in an active HCV cohort of 39 patients composed of 28 patients tested by RNA-seq and 11 randomly selected additional patients with clinical characteristics and tumor stage matched with the post-DAA cohort (as reported in Table 1). FOXM1 and Ki67 were chosen on the basis of gene expression in HCC tissues and treated cell lines (Figure 4F) and for their role in HCC [21,22]. VIM was tested as a marker of epidermal-to-mesenchymal transition (EMT) with molecular relevance in HCC [23]. Both FOXM1 and VIM were upregulated in HCCs from the post-DAA cohort (Figure 5). Remarkably, they were also expressed in the active HCV cohort, not previously exposed to DAA, though at a lower level as shown in representative cases from both groups (Figure 5). Ki67 labeling index (LI) was assessed to evaluate proliferation of DAA-treated and control HCCs. Ki67 LI was heterogeneous in the small DAA cohort—a few cases displayed high Ki67 LI, approaching 100%, while others displayed an LI slightly lower or similar to controls.

Though far away from being conclusive, this preliminary study on a very small DAA-related HCC series, suggested that in vitro studies might be representative of primary DAA-related HCCs.

## 4. Discussion

In this study, we showed evidence that a possible interference of sofosbuvir and daclatasvir with cancer cells exists through off-target effects. This interference might be related to specific cellular backgrounds. This evidence justifies both the occurrence of a particularly aggressive recurrence pattern in some patients and a decreased risk of recurrence in others. Actual DAA indications include chronic hepatitis and liver cirrhosis without HCC, both showing a very low risk of HCC following DAAs, which is reduced more when compared with untreated patients. Conversely, an increased incidence of HCC after DAAs has been reported in high-risk populations such as patients previously treated for HCC, suggesting a possible role in the progression of pre-existing dysplastic/neoplastic foci. The interpretation of clinical findings pointed so far to a reduced immunosurveillance of neoplastic clones because of a rapid decrease in viremia [8]. Our experimental setting cannot explore this aspect. Similarly, we did not assay molecular changes dependent on viral eradication. Instead, we focused on possible off-target effects directly induced by DAA molecules, considering that these events were previously reported for nucleotide reverse transcriptase inhibitors on mitochondrial DNA polymerase gamma, and result from aberrant DNA chain termination and inhibition of exonuclease function and DNA replication [14,24,25]. By interfering with the synthesis of mitochondrial proteins, reduced ATP synthesis and electron leakage enhance cytotoxic, free radical production. While off-target effects associated with the classes of nucleotide/nucleoside inhibitors are mainly related to the interference with the mitochondrial DNA polymerase gamma and, possibly, cellular RNA polymerases, other classes of DAAs mainly affect enzymatic functions. For instance, the NS3 serine protease inhibitor telaprevir was reported to inhibit the skin serine hydrolase CELA1, thus favoring serious skin reactions. Even though there is few information available regarding the effects of DAAs, it is conceivable that different kinds of off-target effects mainly depend on the specific mechanisms of action of each class of DAAs.

After determining that treatment with DAAs can effectively modulate the proliferation and migration capability of some HCC cell lines, we moved to an RNA-seq approach, confirming gene expression changes in line with phenotypic changes. We first focused on the molecular basis for the divergent susceptibility of HCC cell lines to DAAs. The Kyoto Encyclopedia of Genes and Genomes (KEGG) pathway analysis revealed that sofosbuvir-downregulated genes were enriched in multiple biological processes interfering with mitochondrial function. The downregulation of these gene families might account for the molecular basis of reduced proliferation of Huh7 and SNU449 cells upon treatment with sofosbuvir. Indeed, mitochondrial toxicity is a known side effect of nucleoside reverse transcriptase inhibitors [14,24,25].

DAA treatment in HCC cell lines modulated ribosomal genes and pseudogenes, outlining the relevant contribution of ribosomal changes in this setting. These factors participate in the translation of viral transcripts [26], but are also to key cellular processes such as apoptosis, cell cycle, cell proliferation, migration and invasion, ultimately leading to tumorigenesis [27,28]. In line with this observation, the deregulated expression of small nucleolar RNAs (SNORNAs) further supported ribosomal dysfunctions in DAA-treated cells. Indeed, SNORNAs guide endo- and exo-nucleolytic cleavages of rRNA precursors, influencing ribosomal biogenesis, besides splicing and translation of essential viral proteins [29], and their participation in cancer development and progression is known [30]. The modulation of histone-coding genes observed in DAA-treated cells might also contribute to impair replication fork progression and influence DNA transcription [31].

These data do not allow to define the relevance of each molecular event, among those identified here, in a tumorigenic or tumor suppressor perspective. It is conceivable that the molecular background might play a central role in dictating the effective relevance of mitochondrial toxicity, ribosomal alterations and epigenetic or transcriptomic changes in each setting.

To assess whether our in vitro findings might reflect the biology of human HCC, we analyzed the overlapping genes regulated by DAAs and those significantly modulated in HCC compared with matched cirrhosis. In line with the opposite effects induced by sofosbuvir and daclatasvir on cell proliferation, most overlapping genes among those regulated by DAAs and those modulated in HCC went in the opposite direction in the case of sofosbuvir, while they were concordant in the case of daclatasvir treatment. Moreover, common genes involved in cell motility were concordant in HCC tissues and in sofosbuvir-treated cells that displayed increased migration capability.

A further supportive contribution came from IHC and in vitro data displaying respectively an increase of EMT tissue markers such as vimentin and FOXM1 in DAA-related HCCs and enhanced migration capabilities in specific DAA-treated HCC cell lines. These findings let us to suppose that, in specific contexts, DAAs might elicit pro-tumorigenic molecular changes in HCC cells and thus contribute to aggressive HCC features. At the same time, in vitro evidence also supported reduced proliferation in specific cell lines resulting from DAAs, in line with the benefits reported in some clinical studies. Of course, HCCs available for molecular characterization were presumably those in which DAA predominant effects had been characterized by increased proliferation and migration. Unfortunately, the surgical series of HCCs diagnosed or recurred soon after DAA treatments was very small because these HCCs are often diagnosed beyond surgical criteria. Of note, all cases in which beneficial/tumor suppressive effects of DAAs predominate are not likely to develop HCC and thus cannot be studied. 

A major limit of the present study was that, while identifying the molecular background as a relevant player mediating the effects of DAAs, specific drivers were not identified. In addition, this study focused on molecular changes occurring in cancer cells, without any evaluation of tumor microenvironment, which is supposed to play a relevant role too. Clinical trials concordantly point to a reduction of HCC development in patients with chronic hepatitis without cirrhotic evolution or HCC, ruling out oncogenic effects of DAAs on normal hepatocytes. The nearly absent constitutive proliferation and migration capability of normal hepatocytes prevented a reliable evaluation of possible “oncogenic” or “anti-oncogenic” changes in these cells. Indeed, the expected effects of DAAs in terms of proliferation and migration reduction cannot be observed because of the ‘quiescent-like’ state of normal hepatocytes. Thus, for both epidemiological and technical reasons, the experiments were performed on HCC-derived cell lines. Interestingly, HepG2 and Hep3B cells, displaying the most “epithelial” features, did not show any phenotypic change after DAA treatment. Conversely, the most “mesenchymal” HCC cells were more affected by DAAs. Even though our data were still too preliminary to be translated into the clinical practice, they appeared to indicate that “mesenchymal” features might be more affected by these drugs. At the same time, we cannot exclude that these drugs themselves were responsible for triggering EMT in specific molecular backgrounds. Thus, the characterization of resected HCC or dysplastic nodules by EMT markers prior to DAA treatment is a possible further line of research that can be proposed, which might help to identify those cases that have to be more strictly surveilled during and after DAA treatment or in whom DAA administration could be at least temporarily withheld. In this perspective, relevant molecular information might be derived from the characterization of extracellular vesicles, which might provide a source of non-invasive biomarkers to be repeatedly assayed.

## 5. Conclusions

This study added a novel observation on the effects of sofosbuvir and daclatasvir by supporting their modulation of proliferation and migration in some HCC- and CCA-derived cell lines. These phenotypic changes were associated with concurrent variations of gene expression likely resulting from off-target effects. Several gene families modulated in vitro by DAAs were deregulated in primary HCCs too. Thus, while treating HCV infection definitely reduces HCC incidence, a field to investigate further is the possible off-target effects of DAAs, which in some instances may explain HCC risk reduction, while in others might support its increase. Our findings suggested that a molecular characterization of the cured HCC might be performed before DAA treatment, and patients bearing an increased risk of HCC development or recurrence should continue to be strictly surveilled when DAA treatments are to be prescribed.

## Figures and Tables

**Figure 1 cancers-12-02674-f001:**
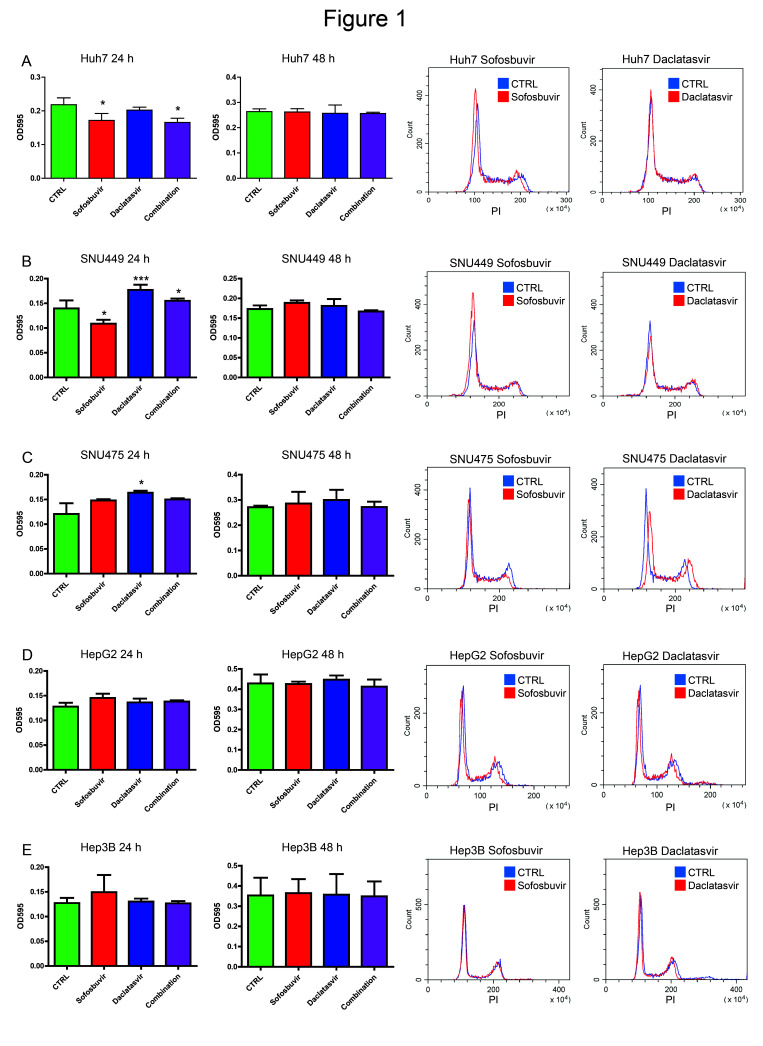
Effects of direct-acting antiviral agents (DAAs) on the proliferation and cell cycle of HCC-derived cell lines. Proliferation assay (24 and 48 h) and cell cycle analysis (24 h) of (**A**) Huh7, (**B**) SNU449, (**C**) SNU475, (**D**) HepG2 and (**E**) Hep3B cell lines following treatment with sofosbuvir (4 µM), daclatasvir (10 nM) and a combination of the two drugs. Control cells (CTRL) were treated with corresponding concentrations of vehicle (dimethyl sulfoxide, DMSO). Student’s *t*-test was used for comparisons. * *p* < 0.05, *** *p* < 0.001. PI: Propidium Iodide.

**Figure 2 cancers-12-02674-f002:**
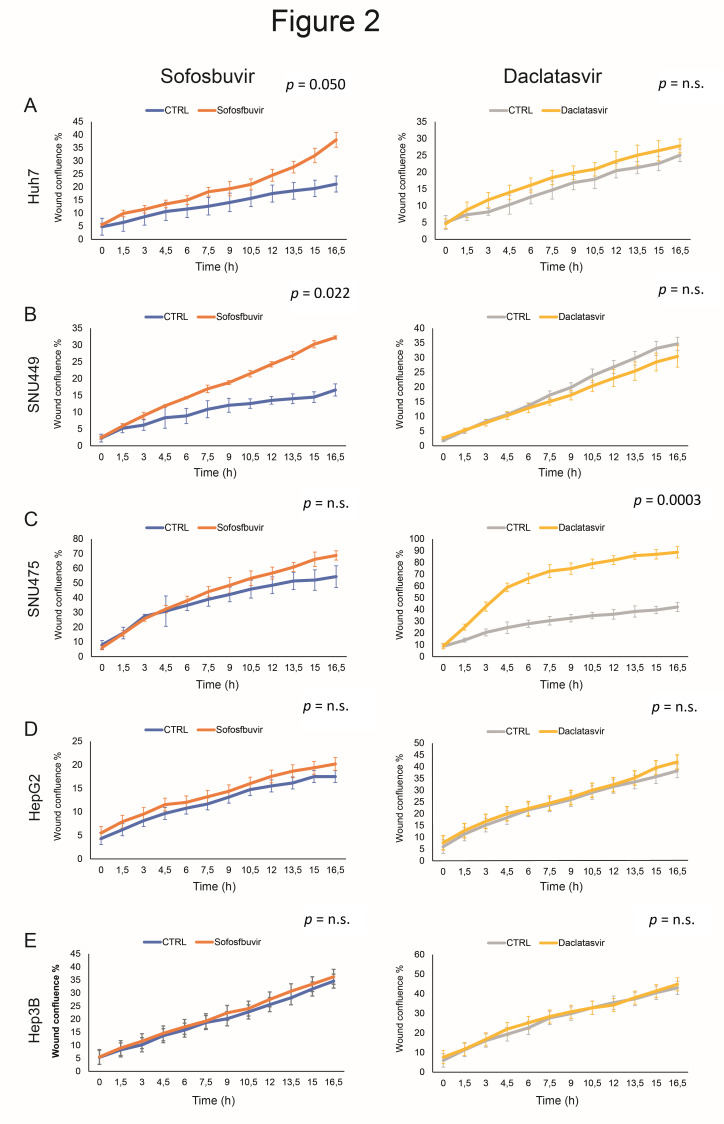
Effects of DAAs on cell migration of HCC-derived cell lines. Cell migration was investigated by the Incucyte Live-Cell Analysis System in (**A**) Huh7, (**B**) SNU449, (**C**) SNU475, (**D**) HepG2 and (**E**) Hep3B cells following treatment with sofosbuvir (4 µM) and daclatasvir (10 nM). Control cells were treated with matched DMSO concentrations, accounting for differences observed in controls from the same cell line. n.s.: not significant.

**Figure 3 cancers-12-02674-f003:**
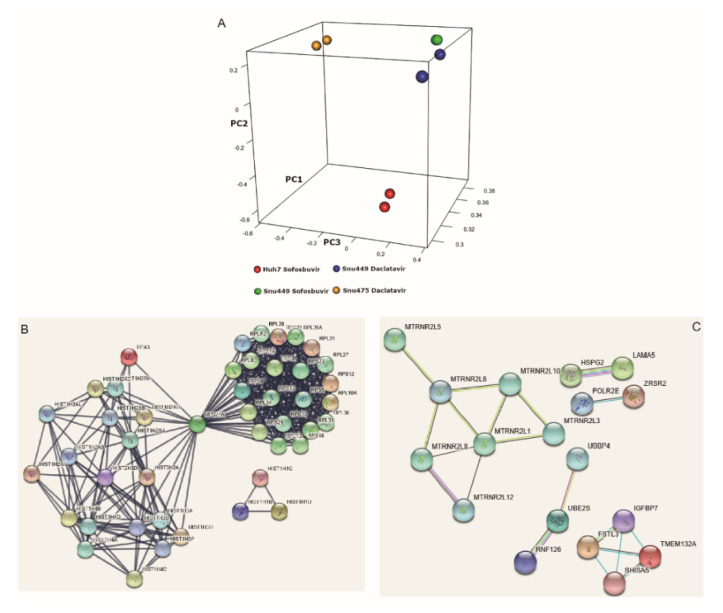
Effects of DAAs on gene expression of HCC-derived cell lines. (**A**) 3D projection of principal component analysis. Separation between different cell lines is visible on both second and third components. (**B**) STRING analysis of commonly downregulated genes in SNU449 and Huh7 treated by sofosbuvir evidenced two major interaction networks: histone- and ribosomal protein-coding genes. (**C**) STRING analysis of genes downregulated by sofosbuvir and upregulated by daclatasvir in SNU449 cells.

**Figure 4 cancers-12-02674-f004:**
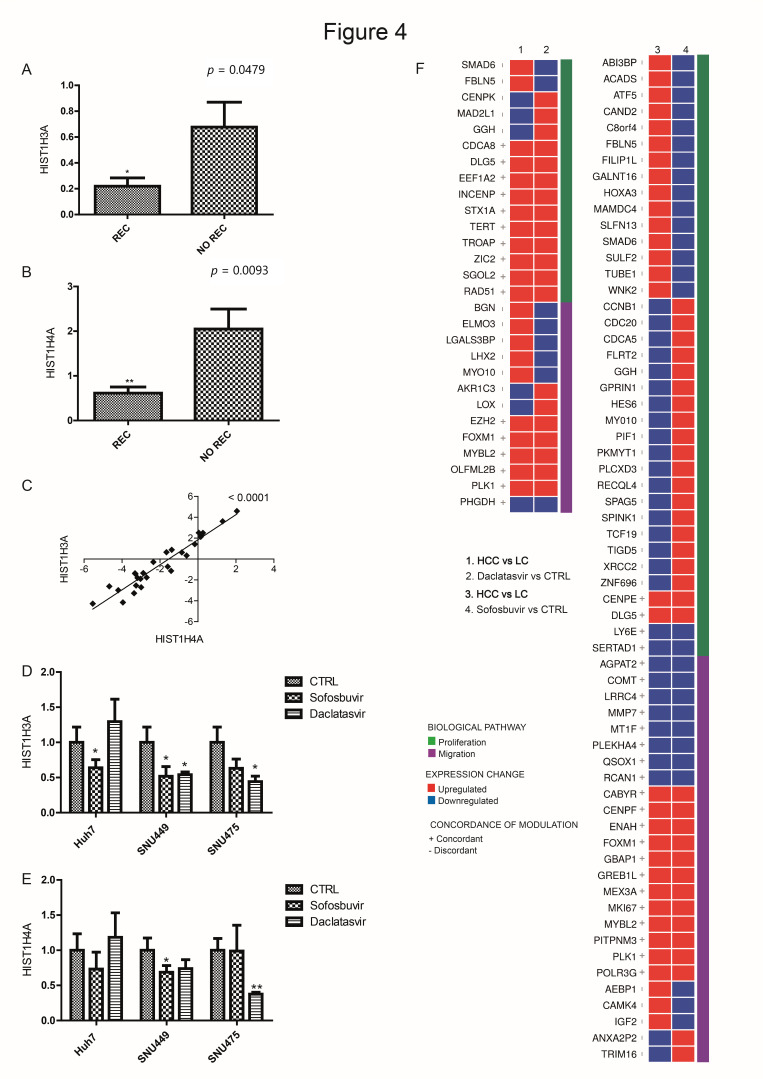
Histone gene expression in DAA-treated cells and primary HCCs. (**A**,**B**) Column graphs of Histone cluster 1 H3 family member A (HIST1H3A) and Histone cluster 1 H4 family member A (HIST1H4A) gene expression in HCCs with (REC) or without (NO REC) tumor recurrence (‘control cohort’). On the top of each graph is reported the *p*-value relative to Student’s *t*-test. Y-axes report 2^(−^^ΔΔCt)^ values. Mean ± SEM values are reported. (**C**) Correlation graph between HIST1H3A and HIST1H4A in HCC tissues from the ‘control cohort’. Y-axes report 2^(−^^ΔΔCt)^ values transformed in a log2 form. *p*-value from Pearson’s correlation is shown on the top of the graph. (**D**,**E**) Column graphs of HIST1H3A and HIST1H4A mRNA expression in DAA-treated HCC cell lines (24 h). Student’s *t*-test was used for comparison between treatment and control (CTRL) cells. Y-axes report 2^(−^^ΔΔCt)^ values. Mean ± SD values are reported. Experiments were repeated twice in triplicate. * *p* < 0.05, ** *p* < 0.01. (**F**) Comparison between deregulated genes in DAA-treated cells and primary HCCs versus matched cirrhotic tissues. Red boxes represent upregulated genes and blue boxes represent downregulated genes, considering HCC versus cirrhotic tissues (upper line of each graph) and treated versus control cell lines (lower line of each graph). Violet bars indicate genes mainly involved in cell motility/migration, while green bars indicate genes mainly involved in cell proliferation. LC: liver cirrhosis.

**Figure 5 cancers-12-02674-f005:**
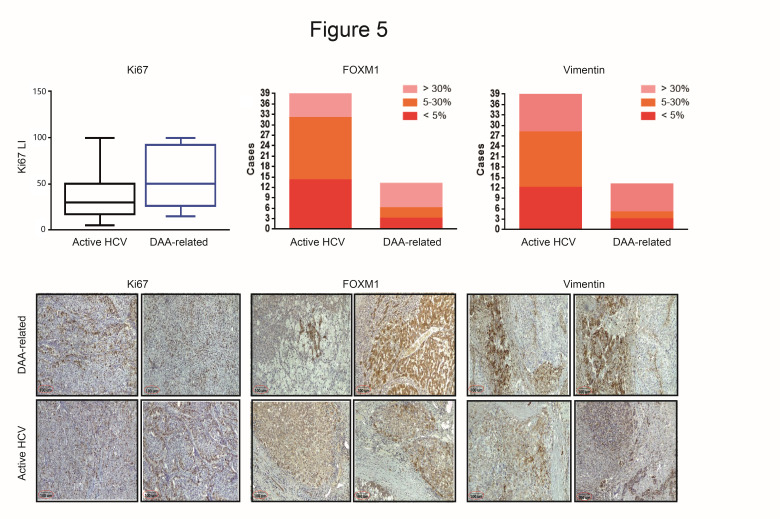
Immunohistochemistry (IHC) evaluation of FOXM1, Ki67 and vimentin (VIM) in HCC tissue. Representative cases of HCCs from the DAA and active HCV patient cohorts. Scale bar: 100 µm.

**Table 1 cancers-12-02674-t001:** Hepatocellular carcinoma (HCC) pathological features.

Clinical Feature	Post-DAA Cohort *n* = 13 (IHC)	Active HCV Cohort *n* = 39 (RNA-Seq, IHC and qPCR)	Difference
Gender (M/F)	8/5	25/14	*p* = n.s.
AFP ^1^	<20 ng/mL 8 (61.5%)	22 (56.4%)	*p* = n.s.
	>20 ng/mL 5 (38.5%)	17 (43.6%)	*p* = n.s.
Primary/recurrence	13/0 (100%)	39/0 (100%)	*p* = n.s.
Maximum HCC size	<5 cm: 12 (92.3%)	<5 cm: 32 (82.1%)	*p* = n.s.
	>5 cm: 1 (7.7%)	>5 cm: 7 (17.9%)	*p* = n.s.
Unifocal	10 (76.9%)	34 (87.2%)	*p* = n.s.
Multifocal	3 (23.1%)	5 (12.8%)	*p* = n.s.
Grading G1-G2	6 (46.1%)	21 (53.8%)	*p* = n.s.
Grading G3-G4	7 (53.8%)	18 (46.2%)	*p* = n.s.
Etiology of CLD ^2^	HCV (cleared): 13 (100%)	HCV (active): 39 (100%)	

All patients gave written informed consent to the study, which was approved by the Ethics Committee of the St. Orsola-Malpighi University Hospital of Bologna (Prot. RNA-seq-HCC; N.138/2015/O/TESS) on 31 July 2015. Histopathological grading was scored according to Edmondson and Steiner criteria. Changes among the two patients’ cohorts are statistically not significant (n.s.). ^1^ Alpha-fetoprotein. ^2^ Chronic liver disease. IHC: immunohistochemistry.

**Table 2 cancers-12-02674-t002:** The top 10 GO-BP terms downregulated in SNU449 and Huh7 by sofosbuvir.

Pathways	*p*-Value	Count
Translation	1.2 × 10^−21^	29
Viral transcription	1.5 × 10^−18^	20
rRNA processing	1.2 × 10^−17^	24
Oxidative phosphorylation	4.8 × 10^−16^	21
Respiratory chain	4.9 × 10^−9^	10
Electron transport	4.9 × 10^−8^	11
ATP biosynthetic process	8.5 × 10^−6^	6
Histone core	6.9 × 10^−5^	8
Anaphase-promoting complex	1.2 × 10^−5^	8
NF-kappa B signaling	4.6 × 10^−5^	7

GO, gene ontology; BP, biological process.

**Table 3 cancers-12-02674-t003:** Overlapping genes downregulated by sofosbuvir and upregulated by daclatasvir in SNU449.

Gene Families	Genes
Mitochondrial DNA-like sequences	*MTRNR2L1*, *MTRNR2L10*, *MTRNR2L12*, *MTRNR2L2*, *MTRNR2L3*, *MTRNR2L5*, *MTRNR2L6*, *MTRNR2L8*
Ribosomal protein (RP)-coding genes and pseudogenes (RPL)	*RP11-329L6.1*, *RP11-36C20.1*, *RP11-75L1.2*, *RP11-832N8.1*, *RP11-889L3.1*, *RP3-375P9.2*, *RP11-1136G11.6*, *RP11-121L10.3*, *RP11-153M3.1*; *RPL13AP20*, *RPL41P2*, *RPL4P5*, *RPL7P47*, *RPS11P5*, *RPS18P9*
Small nuclear RNA genes (RNU) and pseudogenes (RN)	*RNU4-1*, *RNU4-2*, *RNU5B-1*, *RNU5E-1*, *RNU6ATAC*, *U2*; *RN7SKP217*, *RN7SKP255*, *RN7SKP274*, *RN7SKP80*, *RN7SKP9*, *RN7SKP90*, *RN7SKP91*, *RN7SL616P*, *RN7SL704P*, *RNF126*
Ribonucleoproteins	*RNY4*, *RNY4P1*
Small nucleolar RNAs	*SNORA42*, *SCARNA13*
MicroRNAs	*MIR3648*, *MIR3687*
Glyceraldehyde-3-phosphate dehydrogenase pseudogenes	*GAPDHP1*, *GAPDHP38*, *GAPDHP60*
Miscellaneous	*AC004453.8, AC009245.3*, *ACTBP2*, *AURKAIP1*, *BCL2L12*, *C20orf24*, *CHTF18*, *CISD3*, *CTD-2184D3.1*, *DUT*, *ELP5, FSTL3, FTLP3, GDF15, H1FX*, *HNRNPKP4*, *HSF1*, *IGFBP7*, *INPP5E*, *KIFC2*, *LAMA5*, *LDHAP3*, *LMF2*, *MAP1S*, *MED16, NDUFA13, NOTCH1, PCNXL3*, *POLR2E*, *PPIAP29*, *PTOV1*, *REEP4*, *SCAMP4*, *SHISA5*, *SLC25A29*, *TMEM132A*, *TMSB10P1*, *TONSL*, *TTYH3*, *TUBBP1*, *UBBP4*, *UBE2S*, *VKORC1*
Translation elongation factor pseudogenes	*EEF1A1P11*, *EEF1A1P13*, *EEF1A1P25*, *EEF1A1P9*
Heat shock protein family A pseudogenes	*HSP90AB2P*, *HSP90AB3P*, *HSPA8P8*, *HSPG2*
Zinc finger proteins	*ZNF512B*, *ZNF579*

**Table 4 cancers-12-02674-t004:** Overlapping genes deregulated in SNU475 + daclatasvir, SNU449 + sofosbuvir and Huh7 + sofosbuvir (all with increased migration).

Gene Families	Genes
Mitochondrial DNA-like sequences:	MTRNR2L1, MTRNR2L6
Ribosomal protein (RP)-coding genes and pseudogenes (RPL)	*RNA5-8SP2*, *RPL41P2*, *RP11-395B7.7*, *RP5-857K21.11*
Small nuclear RNA genes (RNU)	*RNU5A-1*, *RNU2-6P*, *RNU6ATAC*, *RNY5*, *RNU5A-8P*
Small nucleolar RNAs and pseudogenes	*SNORA54*, *SNORA26*, *SCARNA6*, *SNORA71D*, *Y*, *U2; RN7SKP9*
MicroRNAs	*MIR3648*
Histones	*HIST1H3A*, *HIST1H4D*, *HIST1H4A*, *HIST1H2AB*, *HIST1H2AI*
Miscellaneous	*LAMA5*, *HSPG2*, *ERVK13-1*

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
