# Peer review of "Direct Antiviral Treatments for Hepatitis C Virus Have Off-Target Effects of Oncologic Relevance in Hepatocellular Carcinoma"

_cancers, 2020, doi:10.3390/cancers12092674_

Round 1

Reviewer 1 Report

This is an interesting paper describing functional and molecular alterations in different cell lines upon administration of DAA. Unfortunately no primary cells are analysed. Nevertheless the demonstrated results are interesting and may add important new information in the discussion on whether HCC may be triggered by DAA in patients with HepC. I would only suggest to focus the discussion more on the potential clinical implications of the data.

Author Response

Reviewer 1

This is an interesting paper describing functional and molecular alterations in different cell lines upon administration of DAA. Unfortunately no primary cells are analysed. Nevertheless the demonstrated results are interesting and may add important new information in the discussion on whether HCC may be triggered by DAA in patients with HepC. I would only suggest to focus the discussion more on the potential clinical implications of the data.

R: We thank the Reviewer for this suggestion. At the end of the discussion (page 18, lines 13-23) and at the end of the conclusions (page 18, last 2 lines), we have introduced a possible clinical implication of our results:

“Even though we are well aware that our data are still too preliminary to be translated into the clinical practice, they appear to indicate that “mesenchymal” features are those which might be more affected by these drugs. At the same time, we cannot exclude that these drugs themselves are responsible for triggering EMT in specific molecular backgrounds. Thus, the characterization of resected HCC or dysplastic nodules by EMT markers prior to DAA treatment is a possible further line of research that can be proposed, which might help to identify those cases to be more strictly surveilled on and after DAA treatment or in whom DAA administration could be at least temporarily withheld. In this perspective, relevant molecular information might be derived from the characterization of extra cellular vesicles which might provide a source of non-invasive biomarkers to be repeatedly assayed.” 

Reviewer 2 Report

I have completed my peer review the manuscript “Direct-antiviral treatments for hepatitis C virus have off-target effects of oncologic relevance "[Manuscript Number: cancers-894142]. This document is very interesting, focusing on another effects except anti-viral effect by sofosbuvir and asunaprevir and suggesting a relationship with the hepatocarcinogenic potential that remains after viral treatment. Comments and feedback for the authors can be found below.

(1) For clinical use, indication of direct antiviral agent is chronic hepatitis, liver cirrhosis, and chronic liver disease without HCC. All cell lines used for in vitro analysis this time are liver cancer cell lines. The effects of sofosbuvir and daclatasvir in the non-carcinogenic state, or without HCC state, should be observed using primary cultured hepatocytes.

(2) According to this analysis, there were several off target effects in sofosbuvir and daclatasvir, therefore, it is not controversial to conclude that it may affect carcinogenesis after clearance of the virus.

There are many other drugs in the same class, for example, ladipasvir, asnaprevir, etc. It is speculated that these drugs also have off target effects. Please indicate whether the off target occurs in each class such as NS3/4 serine protease inhibitor or RNA polymerase inhibitor or drug-specific.

(3) Related to the question (2), and also mentioned in the discussion by the author, is it possible to show what kind of off target effect actually contributes most to hepatocarcinogenesis?

Also, when using this information in clinical practice, please give your opinion on the possibility of predicting cancer, for example, by using information in liver tissue or circulating nucleated cell components or plasma after the end of treatment.

(4) The item of 5.5 is duplicated in Result.

Author Response

Reviewer 2

I have completed my peer review the manuscript “Direct-antiviral treatments for hepatitis C virus have off-target effects of oncologic relevance "[Manuscript Number: cancers-894142]. This document is very interesting, focusing on another effects except anti-viral effect by sofosbuvir and asunaprevir and suggesting a relationship with the hepatocarcinogenic potential that remains after viral treatment. Comments and feedback for the authors can be found below.

(1) For clinical use, indication of direct antiviral agent is chronic hepatitis, liver cirrhosis, and chronic liver disease without HCC. All cell lines used for in vitro analysis this time are liver cancer cell lines. The effects of sofosbuvir and daclatasvir in the non-carcinogenic state, or without HCC state, should be observed using primary cultured hepatocytes.

R: we agree with the Reviewer that actual DAA indications are chronic hepatitis, liver cirrhosis, and chronic liver disease without HCC. This is mainly because treatment of HCV in patients with HCC is not likely to be cost effective due to the poor prognosis of active HCC. Patients with cured HCC are instead often considered as without HCC and may be treated. However, it is well known that “cure” of HCC means necrosis of all detectable tumors, but microscopic satellites may remain viable. These patients are probably the ones for whom off-target effects of DAAs are of highest interest. Indeed, clinical studies (most of which cited in the paper) tell us that DAAs reduce neoplastic development in patients without previous experience of HCC. Patients with chronic hepatitis show a very low risk of HCC following DAAs, which is even reduced when compared to untreated patients. Thus, we can affirm that clinical studies do not support any oncogenic effect of DAAs on non-neoplastic hepatocytes and, for this reason, we think it is not appropriate to test such hypothesis, not supported by clinical observation, which instead point at patients who already bear HCC cells in the liver at a microscopic level. We thank the Reviewer for raising this point, which was added in the discussion (page 16, lines 6-11) to better explain our experimental design and our choices for in vitro experiments.

A further point, that discouraged us from pursuing the Reviewer’s suggestion, is that normal hepatocytes display a very low proliferation and migration capability. These characteristics hamper the possibility to reveal both oncogenic and anti-oncogenic functions of short/intermediate-term (24- or 48-hours) treatments such as those performed with DAAs. Especially, in our opinion, normal hepatocytes do not represent a good in vitro model to look at possible anti-oncogenic functions due to their inability to proliferate and migrate.  Indeed, to our knowledge, primary hepatocytes are suggested for metabolism studies, hepato-toxicity tests, drug-drug interaction assays, specific genotoxic tests, but their proliferation and migration capability in cultures is really limited and thus they are not suitable for proliferation assays, cell cycle analysis or migration assays such as those performed in our study. These affirmations are supported by several studies and reviews, one of the many by Guo X et al (Journal of Toxicology and Environmental Health -  critical reviews. 2020, vol 23, N.1: 27-50) which also warns about early phenotypic changes of primary hepatocytes when cultured, together with a relevant inter-donor variability and lack of proliferative capacity. A sentence has been added to the discussion in order to make our choice clearer to the reader too (discussion, page 17, lines 3-10).

Nonetheless, we tested primary hepatocytes (Hep10) and we confirmed their very low proliferation and nearly absent migration capability. This further confirmed that beneficial effects of DAAs reported in patients with chronic hepatitis without HCC could not be evaluated in non-proliferating cell cultures, thus a relevant piece of information would be unavailable (e.g. the anti-oncogenic effect of DAAs in patients without HCC as suggested by clinical observation), making experiments unbalanced and misleading.

(2) According to this analysis, there were several off target effects in sofosbuvir and daclatasvir, therefore, it is not controversial to conclude that it may affect carcinogenesis after clearance of the virus.

R: Our study was addressed to explore possible off-target effects of DAAs, independently of HCV presence or clearance. We thank the Reviewer for this elucidation, which has been introduced in the discussion (page 15 lines 13-15). We have now remarked that the study was focused on possible off-target effects directly induced by DAA molecules.

There are many other drugs in the same class, for example, ladipasvir, asnaprevir, etc. It is speculated that these drugs also have off target effects. Please indicate whether the off target occurs in each class such as NS3/4 serine protease inhibitor or RNA polymerase inhibitor or drug-specific.

R: As stated in the introduction, we have focused on Sofosbuvir and daclatasvir because they have been the mostly used DAAs in Italian cirrhotic patients. Indeed, we aimed to uncover any possible explanation for what we directly observed in our clinical practice. Remarkably, the even small series of patients we tested here, was treated with these two specific drugs. Concerning other drugs belonging to the same class of daclatasvir, as observed by the Reviewer, the literature tells us that among NS5A inhibitors, daclatasvir and ledipasvir deserve attention as inducers of a dose-dependent DNA damage through the oxidation of both nucleobases and deoxyribose moieties of DNA (El-Yazbi AF, Loppnow GR. Probing DNA damage induced by common antiviral agents using multiple analytical techniques. Journal of Pharmaceutical and biomedical analysis 157 (2018): 226-234). An experimental approach different from our was used in that study, thus we think the results are hard to be compared, however these data were reported in the introduction (page 5, last line and page 6, first 2 lines).

Concerning other classes of DAAs, clinical side effects of Telaprevir, a HCV-NS3 serine protease inhibitor, were correlated with off-target effects. More specifically, the serious skin reactions sometimes induced by telaprevir were suggested to result from an off-target inhibition of CELA1, a serine hydrolase expressed in the skin (Bachovchin Da et al. Nat Chem Biol 2014: 10 (4): 656-663). We thank the Reviewer for this suggestion. The citation of these previous findings in the introduction (page 5, lines 28-33) makes our working hypothesis more reliable. Again, our experiments do not allow us to speculate on off-target effects of other classes of direct anti-viral agents. Remarkably, as reported by Bachovchin et al, the off-target effect is not class dependent, but drug-specific and structurally similar drugs of the same class do not necessarily share the same off-target effect. This observation further prevents us to generalize results obtained with a specific drug to its whole class.

(3) Related to the question (2), and also mentioned in the discussion by the author, is it possible to show what kind of off target effect actually contributes most to hepatocarcinogenesis?

R: actually it is very hard to identify which kind of off-target effect contribute most to hepatocarcinogenesis. From a molecular point of view, the most relevant and common transcriptomic changes involve mitochondrial and ribosomal functions and histone core composition, as described in the original version of the paper. It is not possible to state which of these might play a driver function. We thank the Reviewer for this suggestion and we have added a sentence in the discussion (page 17, lines 7-11): “These data do not allow to define the relevance of each molecular event among those identified here, in a tumorigenic or tumor suppressor perspective. It is conceivable that the molecular background might play a central role in dictating the effective relevance of mitochondrial toxicity, ribosomal alterations, epigenetic or transcriptomic changes in each setting”.

In addition, taking advantage from DAA-induced changes in different cell lines, we also speculated that the “mesenchymal” molecular backgrounds might be more affected by these class of drugs. Thus, the characterization of resected HCC or DN for EMT molecules prior to DAA treatment is a possible further line of research that can be proposed, which might help to identify those cases to be more strictly surveilled on and after DAA treatment. A sentence has been added at the end of the Discussion (page 17, lines 13-20).  

Also, when using this information in clinical practice, please give your opinion on the possibility of predicting cancer, for example, by using information in liver tissue or circulating nucleated cell components or plasma after the end of treatment.

R: We have no data on circulating cells or components of plasma, however, it is conceivable that relevant information might be obtained by circulating extra cellular vesicles. We thank the Reviewer for this suggestion and we have added a sentence to the discussion (page 18, lines 21-23): “In this perspective, relevant molecular information might be derived from the characterization of extra cellular vesicles which might provide a source of non-invasive biomarkers to be repeatedly assayed.”

 (4) The item of 5.5 is duplicated in Result.

R: according to Reviewer 4, the imaging study was deleted.

Reviewer 3 Report

The authors evaluated the changes of proliferation and migration capability and gene expression profile in HCC cell lines after sofosbuvir or daclatasvir treatment, and found that proliferation and migration capability was affected by these drugs, dependent on cellular backgrounds, in association with concurrent variations of gene expression. In addition, several gene families modulated by these drugs in vitro were also deregulated in primary HCC, and DAA-treated HCC had high proliferative index and EMT property. The authors concluded that DAA has off-target effects modulating aggressive behaviors in HCC.

This study was well done and the manuscript is well-written, although clinical validation was not enough.

  1. In Figure 5, the authors presented the results of immunohistochemical Foxm1 and vimentin expression by box plot. However, these expressions were assessed by semequatitative manner, so the results should be shown by the number of cases in each scale.

Author Response

Reviewer 3

The authors evaluated the changes of proliferation and migration capability and gene expression profile in HCC cell lines after sofosbuvir or daclatasvir treatment, and found that proliferation and migration capability was affected by these drugs, dependent on cellular backgrounds, in association with concurrent variations of gene expression. In addition, several gene families modulated by these drugs in vitro were also deregulated in primary HCC, and DAA-treated HCC had high proliferative index and EMT property. The authors concluded that DAA has off-target effects modulating aggressive behaviors in HCC.

This study was well done and the manuscript is well-written, although clinical validation was not enough.

R: We agree with the Reviewer that the clinical validation of in vitro findings is very preliminary and not conclusive, as stated in the original manuscript and further outlined in the amended manuscript. To this end, a sentence justifying the small number of DAA-related HCCs has been added to the discussion (page 16, lines 28-30). Unfortunately, in ours as well as in other centers, HCCs diagnosed or recurred after DAA treatments are rarely treatable by surgery, because they are diagnosed beyond surgical criteria. Thus, gaining a numerous series of tissues to be fully characterized is really difficult.

  1. In Figure 5, the authors presented the results of immunohistochemical Foxm1 and vimentin expression by box plot. However, these expressions were assessed by semequatitative manner, so the results should be shown by the number of cases in each scale.

R: Graphical representation of FOXM1 and VIM has been changed according to the Reviewer’s suggestion. Now cases have been grouped according to the percentage of positive cells: <5%, 5-30% and >30% of positive cells are visualized by different colors. Ki67 LI is still represented by box-plot since it is a continuous variable. 

Reviewer 4 Report

  1. In the title I suggest to add at the end    … in HCC.
  2. Please, in the abstract, add, also in extention: Epithelial-Mesenchymal Transition (EMT).
  3. The paper concerns on two sections. The first is a main, well structured in vitro experimental section, that describes cell proliferation and migration assays in DAA exposed cell coltures, showing a clear influence in some HCC-derived cell lines. In the second part of the study try to correlate the results obtained in vitro with some in vivo features, particularly studing the IHC expression of some tissue markers relevant in HCC. Three cohort are introduced: 1) the DAA cohort (13 cases, HCV cleared after DAA), 2) the control cohort (26 cases, 11HCV active+15 NASH) and3) the NGS cohort (28 cases, non exposed to DAA, HCV active.However these 3 cohorts are heterogeneous and were shown in a confusing way in table 2 of supplementary material.
  4. I suggest to review the case series and to compare only 2 groups: 1) the DAA cohort versus 2) a control group of only HCV active cases (28 of NGS cohort and 11 of control cohort), thus to obtain a comparison of cases exposed vs not exposed to DAA, describing also the p levels of the parameters considered.
  5. Supplementary table 2, as proposed and modified with only two groups and without imaging criteria may be moved into the paper.
  6. I think that the part of the table regarding the imaging criteria is inadequate, as it is not presented in material and methods and so it is difficult to understand by readers and probably it may be removed.
  7. Thus also consider to erase paragraph 5.5. Imaging characteristics of HCC.... These data are weak and do not supported by adequate figures.
  8. This new presentation should modify also figure 5.
  9. This type of proposed comparison could be more relevant , as it will made between cleared and viraemic cases, and could probably permit some speculations about one of the questions that the authors supposed (ie if the sudden eradication can cause an immunological imbalance triggering HCC).
  10. Please modify Figure 2, on a black background is hard to read.
  11. Supplementary Figure 1, the part E in black background is not visible and in the last line of the caption there is an error (E-H) ??
  12. Table 1 may be can move to page 9 after the citation in the text Table 1
  13. Figures 3B and 3D may be magnified and moved to the supplementary section.
  14. Figure 4F may be magnified and turn in vertical manner to be visible.
  15. Please introduce a new paragraph on IHC description: 5.6 Immunohistochemistry evaluation in HCC with and without DAA.
  16. … from “To further assess the estent to which in vitro.... on page 14.

Author Response

Reviewer 4

  1. In the title I suggest to add at the end  … in HCC.

R: we thank the Reviewer for this suggestion

  1. Please, in the abstract, add, also in extention: Epithelial-Mesenchymal Transition (EMT).

R: we thank the Reviewer for this suggestion

  1. The paper concerns on two sections. The first is a main, well structured in vitro experimental section, that describes cell proliferation and migration assays in DAA exposed cell coltures, showing a clear influence in some HCC-derived cell lines. In the second part of the study try to correlate the results obtained in vitro with some in vivo features, particularly studing the IHC expression of some tissue markers relevant in HCC. Three cohort are introduced: 1) the DAA cohort (13 cases, HCV cleared after DAA), 2) the control cohort (26 cases, 11HCV active+15 NASH) and3) the NGS cohort (28 cases, non exposed to DAA, HCV active. However these 3 cohorts are heterogeneous and were shown in a confusing way in table 2 of supplementary material.

  1. I suggest to review the case series and to compare only 2 groups: 1) the DAA cohort versus 2) a control group of only HCV active cases (28 of NGS cohort and 11 of control cohort), thus to obtain a comparison of cases exposed vs not exposed to DAA, describing also the p levels of the parameters considered.

R: The control group, made of both active HCV infection and NASH patients, was selected by considering: 1) that cured HCV patients might be more similar to NASH than to active HCV patients, given the common finding of steatosis in HCV infected patients and, 2) to reduce the known molecular effects in terms of transcriptional changes, induced by active HCV infection in liver tissue. By including only a subgroup of such active HCV infected patients and by considering mean and median transcriptomic changes in the whole series of “control” HCC tissues (active HCV and NASH) we aimed to reduce the effects of active HCV infection, while still having the effects of steatosis-induced changes. However, we have not confirmed these assumptions by proper molecular studies in our patients cohorts, and we recognize that these assumptions may be somehow arbitrary, despite some published studies support them. Thus, we have accepted the suggestion of the Reviewer and in the amended version of the study we have compared HCC tissues from HCV treated patients with HCC tissues from active HCV infected patients.

As remarked in the description of patients in the material and methods section (completely re-written), the new control group of 39 patients was analysed by IHC and RT-PCR, but only 28 of them were tested by RNA-seq. In the paragraph “Immunohistochemistry evaluation in HCC with and    without DAAs” (results section), we introduced the novel “active HCV cohort” of 39 patients composed of 28 patients tested also by RNAseq and eleven additional patients with clinical characteristics and tumor stage matched with the post-DAA cohort (page 14, lines 21-24).

  1. Supplementary table 2, as proposed and modified with only two groups and without imaging criteria may be moved into the paper.

R: Supplementary Table 2, now Table 1, has been moved into the paper and two groups only are now displayed: DAA-treated patients with cleared HCV infection and patients with active HCV infection.

  1. I think that the part of the table regarding the imaging criteria is inadequate, as it is not presented in material and methods and so it is difficult to understand by readers and probably it may be removed.

R: imaging characteristics of HCC have been erased from Supplementary Table 2 (now Table 1) and from the paper.

  1. Thus also consider to erase paragraph 5.5. Imaging characteristics of HCC.... These data are weak and do not supported by adequate figures.

R: imaging characteristics of HCC have been erased from the paper as well as from Supplementary Table 2 (now Table 1).

  1. This new presentation should modify also figure 5.

R: Figure 5 has been modified with the new results of Ki67 LI and FOXM1 and VIM IHC assays in the new control cohort (the 13 patients DAA-cohort is unchanged). The graphical representation of FOXM1 and VIM data was modified, according to the suggestion of Reviewer 3.

  1. This type of proposed comparison could be more relevant , as it will made between cleared and viraemic cases, and could probably permit some speculations about one of the questions that the authors supposed (ie if the sudden eradication can cause an immunological imbalance triggering HCC).

R: the comparisons have been modified according to the Reviewer suggestion: the 13 HCCs developed after DAA eradication of HCV were compared with a 39 patients cohort composed of 28 HCCs with active HCV infection (the original “RNAseq” cohort) and eleven patients with active HCV infection, derived from the original “control cohort”, deprived of the NASH-related HCCs. Now HCC with active HCV infection are compared with HCC with treated HCV infection.

We would like to remark that our working hypothesis was not to reveal an immunological imbalance after HCV eradication. As stated in the discussion of the original paper, our experimental setting does not allow such speculation on the immunological background. In addition, in the original study, we chose a large proportion of NASH patients in the control cohort, to avoid the impact of active HCV infection on possible transcriptional alterations in infected cells. However we have amended our manuscript according to the Reviewer suggestions, because we are well aware that different point of view in this specific field deserve great attention, and each point of view is worthy to be explored.

  1. Please modify Figure 2, on a black background is hard to read.

  1. Supplementary Figure 1, the part E in black background is not visible and in the last line of the caption there is an error (E-H) ??

R: 10 and 11: R: We thank the Reviewer for this suggestion and we modified the background of graphs in Figure 2 and Supplementary Figure 1E accordingly. The error in the caption has been corrected.

  1. Table 1 may be can move to page 9 after the citation in the text Table 1

R: We agree that Table 1 (now Table 2) should be printed close to its citation in the text.

  1. Figures 3B and 3D may be magnified and moved to the supplementary section.

            R: Figure 3B is now supplementary Figure 2, Figure 3D is now Supplementary Figure 4. Both              have been magnified.

  1. Figure 4F may be magnified and turn in vertical manner to be visible.

R: Figure 4F has been magnified and turned in vertical manner.

  1. Please introduce a new paragraph on IHC description: 5.6 Immunohistochemistry evaluation in HCC with and without DAA.… from “To further assess the estent to which in vitro.... on page 14.

R: A new paragraph has been added in the Results section and the description of IHC assay in the supplementary section has been further detailed.

We thank the Reviewer for the careful revision that allowed us to improve our paper. 

Round 2

Reviewer 2 Report

First I list the original my question, the author's reply, and then a comment on the author's reply.

(1) For clinical use, indication of direct antiviral agent is chronic hepatitis, liver cirrhosis, and chronic liver disease without HCC. All cell lines used for in vitro analysis this time are liver cancer cell lines. The effects of sofosbuvir and daclatasvir in the non-carcinogenic state, or without HCC state, should be observed using primary cultured hepatocytes.

R: we agree with the Reviewer that actual DAA indications are chronic hepatitis, liver cirrhosis, and chronic liver disease without HCC. This is mainly because treatment of HCV in patients with HCC is not likely to be cost effective due to the poor prognosis of active HCC. Patients with cured HCC are instead often considered as without HCC and may be treated. However, it is well known that “cure” of HCC means necrosis of all detectable tumors, but microscopic satellites may remain viable. These patients are probably the ones for whom off-target effects of DAAs are of highest interest. Indeed, clinical studies (most of which cited in the paper) tell us that DAAs reduce neoplastic development in patients without previous experience of HCC. Patients with chronic hepatitis show a very low risk of HCC following DAAs, which is even reduced when compared to untreated patients. Thus, we can affirm that clinical studies do not support any oncogenic effect of DAAs on non-neoplastic hepatocytes and, for this reason, we think it is not appropriate to test such hypothesis, not supported by clinical observation, which instead point at patients who already bear HCC cells in the liver at a microscopic level. We thank the Reviewer for raising this point, which was added in the discussion (page 16, lines 6-11) to better explain our experimental design and our choices for in vitro experiments. A further point, that discouraged us from pursuing the Reviewer’s suggestion, is that normal hepatocytes display a very low proliferation and migration capability. These characteristics hamper the possibility to reveal both oncogenic and anti-oncogenic functions of short/intermediate-term (24- or 48-hours) treatments such as those performed with DAAs. Especially, in our opinion, normal hepatocytes do not represent a good in vitro model to look at possible anti-oncogenic functions due to their inability to proliferate and migrate. Indeed, to our knowledge, primary hepatocytes are suggested for metabolism studies, hepato-toxicity tests, drug-drug interaction assays, specific genotoxic tests, but their proliferation and migration capability in cultures is really limited and thus they are not suitable for proliferation assays, cell cycle analysis or migration assays such as those performed in our study. These affirmations are supported by several studies and reviews, one of the many by Guo X et al (Journal of Toxicology and Environmental Health - critical reviews. 2020, vol 23, N.1: 27-50) which also warns about early phenotypic changes of primary hepatocytes when cultured, together with a relevant interdonor variability and lack of proliferative capacity. A sentence has been added to the discussion in order to make our choice clearer to the reader too (discussion, page 17, lines 3-10). Nonetheless, we tested primary hepatocytes (Hep10) and we confirmed their very low proliferation and nearly absent migration capability. This further confirmed that beneficial effects of DAAs reported in patients with chronic hepatitis without HCC could not be evaluated in non-proliferating cell cultures, thus a relevant piece of information would be unavailable (e.g. the anti-oncogenic effect of DAAs in patients without HCC as suggested by clinical observation), making experiments unbalanced and misleading.

>>The question I would like to ask is that DAA is basically used in the absence of hepatocellular carcinoma, for example, chronic hepatitis, liver cirrhosis or the cases after curative treatment of HCC,  and thus in the absence of cancer cells. I understand the situation of in vitro study, easy and low cost handling of hepatocellular carcinoma-derived cell lines and relatively high cost and low proliferation of primary hepatocyte. No matter how much the hepatocellular carcinoma dirived cell lines has the character of hepatocyte, however, these cell lines are cancer cell. Considering these points comprehensively, I would like you to show the pharmacological effect of DAA in primary culture.

(2) According to this analysis, there were several off target effects in sofosbuvir and daclatasvir, therefore, it is not controversial to conclude that it may affect carcinogenesis after clearance of the virus.

R: Our study was addressed to explore possible offtarget effects of DAAs, independently of HCV presence or clearance. We thank the Reviewer for this elucidation, which has been introduced in the discussion (page 15 lines 13-15). We have now remarked that the study was focused on possible offtarget effects directly induced by DAA molecules.

There are many other drugs in the same class, for example, ladipasvir, asnaprevir, etc. It is speculated that these drugs also have off target effects. Please indicate whether the off target occurs in each class such as NS3/4 serine protease inhibitor or RNA polymerase inhibitor or drug-specific.

R: As stated in the introduction, we have focused on Sofosbuvir and daclatasvir because they have been the mostly used DAAs in Italian cirrhotic patients. Indeed, we aimed to uncover any possible explanation for what we directly observed in our clinical practice. Remarkably, the even small series of patients we tested here, was treated with these two specific drugs. Concerning other drugs belonging to the same class of daclatasvir, as observed by the Reviewer, the literature tells us that among NS5A inhibitors, daclatasvir and ledipasvir deserve attention as inducers of a dose-dependent DNA damage through the oxidation of both nucleobases and deoxyribose moieties of DNA (El-Yazbi AF, Loppnow GR. Probing DNA damage induced by common antiviral agents using multiple analytical techniques. Journal of Pharmaceutical and biomedical analysis 157 (2018): 226-234). An experimental approach different from our was used in that study, thus we think the results are hard to be compared, however these data were reported in the introduction (page 5, last line and page 6, first 2 lines). Concerning other classes of DAAs, clinical side effects of Telaprevir, a HCV-NS3 serine protease inhibitor, were correlated with off-target effects. More specifically, the serious skin reactions sometimes induced by telaprevir were suggested to result from an off-target inhibition of CELA1, a serine hydrolase expressed in the skin (Bachovchin Da et al. Nat Chem Biol 2014: 10 (4): 656-663). We thank the Reviewer for this suggestion. The citation of these previous findings in the introduction (page 5, lines 28-33) makes our working hypothesis more reliable. Again, our experiments do not allow us to speculate on off-target effects of other classes of direct anti-viral agents. Remarkably, as reported by Bachovchin et al, the offtarget effect is not class dependent, but drug-specific and structurally similar drugs of the same class do not necessarily share the same off-target effect. This observation further prevents us to generalize results obtained with a specific drug to its whole class.

>>The use of Daclatasvir and Lady Passville for cirrhosis in Italy helps us understand the background of this study. Also, if I do similar research, I think that drug selection method will be that way. However, since hapatocarcinogenesis after HCV clearance by DAA is the focus of doctors and researchers using DAA, I would like to request general discussion. In this regard, I would like to a clear answer to the text below is provided.

>There are many other drugs in the same class, for example, ladipasvir, asnaprevir, etc. It is speculated that these drugs also have off target effects.Please indicate whether the off target occurs in each class such as NS3/4 serine protease inhibitor or RNA polymerase inhibitor or drug-specific.<

(3) Related to the question (2), and also mentioned in the discussion by the author, is it possible to show what kind of off target effect actually contributes most to hepatocarcinogenesis?

R: actually it is very hard to identify which kind of offtarget effect contribute most to hepatocarcinogenesis. From a molecular point of view, the most relevant and common transcriptomic changes involve mitochondrial and ribosomal functions and histone core composition, as described in the original version of the paper. It is not possible to state which of these might play a driver function. We thank the Reviewer for this suggestion and we have added a sentence in the discussion (page 17, lines 7-11): “These data do not allow to define the relevance of each molecular event among those identified here, in a tumorigenic or tumor suppressor perspective. It is conceivable that the molecular background might play a central role in dictating the effective relevance of mitochondrial toxicity, ribosomal alterations, epigenetic or transcriptomic changes in each setting”. In addition, taking advantage from DAA-induced changes in different cell lines, we also speculated that the “mesenchymal” molecular backgrounds might be unless otherwise stated more affected by these class of drugs. Thus, the characterization of resected HCC or DN for EMT molecules prior to DAA treatment is a possible further line of research that can be proposed, which might help to identify those cases to be more strictly surveilled on and after DAA treatment. A sentence has been added at the end of the Discussion (page 17, lines 13-20).

(3) Related to the question (2), and also mentioned in the discussion by the author, is it possible to show what kind of off target effect actually contributes most to hepatocarcinogenesis?

R: actually it is very hard to identify which kind of offtarget effect contribute most to hepatocarcinogenesis. From a molecular point of view, the most relevant and common transcriptomic changes involve mitochondrial and ribosomal functions and histone core composition, as described in the original version of the paper. It is not possible to state which of these might play a driver function. We thank the Reviewer for this suggestion and we have added a sentence in the discussion (page 17, lines 7-11): “These data do not allow to define the relevance of each molecular event among those identified here, in a tumorigenic or tumor suppressor perspective. It is conceivable that the molecular background might play a central role in dictating the effective relevance of mitochondrial toxicity, ribosomal alterations, epigenetic or transcriptomic changes in each setting”. In addition, taking advantage from DAA-induced changes in different cell lines, we also speculated that the “mesenchymal” molecular backgrounds might be unless otherwise stated more affected by these class of drugs. Thus, the characterization of resected HCC or DN for EMT molecules prior to DAA treatment is a possible further line of research that can be proposed, which might help to identify those cases to be more strictly surveilled on and after DAA treatment. A sentence has been added at the end of the Discussion (page 17, lines 13-20).

>> Factors that contribute to carcinogenesis when DAA is used in patients with cirrhosis, which has high carcinogenic potential, and in patients with cirrhosis after hepatocellular carcinoma treatment, will continue to be requested for analysis.

Also, when using this information in clinical practice, please give your opinion on the possibility of predicting cancer, for example, by using information in liver tissue or circulating nucleated cell components or plasma after the end of treatment.

R: We have no data on circulating cells or components of plasma, however, it is conceivable that relevant information might be obtained by circulating extra cellular vesicles. We thank the Reviewer for this

suggestion and we have added a sentence to the discussion (page 18, lines 21-23): “In this perspective, relevant molecular information might be derived from the characterization of extra cellular vesicles which might provide a source of non-invasive biomarkers to be repeatedly assayed.”

>>There is no objection to this point.

Reviewer 4 Report

I approve all the changes to the paper.

Author Response

We thank the Reviewer for his/her valuable advices.